# A Sensor and Machine Learning-Based Sensory Management Recommendation System for Children with Autism Spectrum Disorders [note 1]

**DOI:** 10.3390/s22155803

**Published:** 2022-08-03

**Authors:** Lingling Deng, Prapa Rattadilok

**Affiliations:** 1School of Computer Science, University of Nottingham Ningbo China, Ningbo 315100, China; 2University of Hertfordshire, Hatfield AL10 9AB, UK; p.rattadilok@herts.ac.uk

**Keywords:** assistive technology, autism spectrum disorders, sensors, wearables, sensory management, machine learning, fuzzy logic

## Abstract

Sensory processing issues are one of the most common issues observed in autism spectrum disorders (ASD). Technologies that could address the issue serve a more and more important role in interventions for ASD individuals nowadays. In this study, a sensory management recommendation system was developed and tested to help ASD children deal with atypical sensory responses in class. The system employed sensor fusion and machine learning techniques to identify distractions, anxious situations, and the potential causes of these in the surroundings. Another novelty of the system included a sensory management strategy making a module based on fuzzy logic, which generated alerts to inform teachers and caregivers about children’s states and risky environmental factors. Sensory management strategies were recommended to help improve children’s attention or calm children down. The evaluation results suggested that the use of the system had a positive impact on children’s performance and its design was user-friendly. The sensory management recommendation system could work as an intelligent companion for ASD children that helps with their in-class performance by recommending management strategies in relation to the real-time information about the children’s environment.

## 1. Introduction

Autism spectrum disorders (ASDs) refer to a group of neurodevelopmental disabilities that affect an individual’s social interactions, communication, interests, and behavior [1]. Sensory processing issue is one of the most common issues observed in individuals with ASD [2]. As many as 90% of ASD individuals may have experienced atypical sensory responses to auditory, visual, touching, tasting, and smelling stimuli [3]. According to the Diagnostic and Statistical Manual of Mental Disorders [2], atypical sensory responses in ASD involve hyper- or hypo-sensitivity to sensory input. ASD individuals who are hypo-sensitive may fail to notice sensory stimuli which typically developing (TD) people can easily detect, resulting in behavioral outcomes such as having difficulty paying attention. Conversely, those who are hyper-sensitive are likely to experience distress to sensory stimuli [4]. Therefore, for ASD individuals with atypical sensory responses, any of the senses at any random time could become hyper-sensitive or hypo-sensitive, which could further trigger discomfort, stress, or distraction [5,6]. ASD individuals could have more problems in sensory management, such as attention and stress management, than their TD peers when processing everyday sensory information [4,7,8]. The sensory issue presents a significant challenge for ASD children and their caregivers, especially within a classroom setting, as it affects their learning efficiency and behavioral control.

The rapid development of information technology has encouraged many researchers to explore technology-based interventions that can be used in daily life for ASD children with atypical sensory responses. Some causes of the above-mentioned discomfort, stress, or distraction in ASD individuals could be found as being part of an ‘unfriendly environment’ [9] (page 92), such as noisy or sudden sounds, bright or dark lights, and warm or cold weather [10,11]. The identification of a triggering event for atypical sensory responses in ASD by technologies could be an important aspect to optimize the surrounding causes related to explosive behaviors in ASD. Traditionally, to measure sensory responses, various manual methods, including rating scales and questionnaires, were used in some systems for predicting the comfort level based on sensory inputs or triggers [8,12]. For instance, Mauro et al. [13] proposed a personalized recommendation system to predict points of interest for individuals with ASD using a Top-N model. Their system used a self-defined sensory profiling questionnaire to obtain information about sensory aversion and preference of people with ASD, generating suggestions for places that are expected to be comfortable and safe for the users. However, even in a safe place, unpleasant sensory factors around the child could appear. To provide effective intervention, exact and real-time identification of sensory causes could be required, which was difficult to achieve merely by manual methods [9].

Recent advancements in sensing technologies have made the sensors work similarly to human senses [9]. Developments in sensor fusion and artificial intelligence have driven the integration of multiple sensors and machine learning for capturing ASD individuals’ real-time sensation in different environments. For instance, a stress-monitoring system for people with ASD was created by Tomczak et al. [14] using low-power wearable sensors. The detection module of the system was built using a heuristic rule-based model. Heart rate, skin conductivity, body temperature, and hand movements were the main indicators to identify a person’s stress responses. Coronato et al. [15] used wearable accelerometers to create a situation-aware system that can identify abnormal motor behaviors. The employment of a neural network model led to an accuracy close to 92% on anomalous gestures of an individual with ASD. Additionally, the internet-connected smart devices had also demonstrated their potential by reducing the reliance on dimensions for monitoring systems [9]. Sula et al. [16] designed a system based on the Internet-of-Things (IoT) that could monitor the body movement and the sensory environment of children with ASD. It could send information about the children’s state in real time to therapists and caregivers using Peer to Peer (P2P) technology. Khullar et al. [9] designed another IoT-based system to detect the environmental information and process the information using fuzzy logic. The system was also able to generate alerts to caregivers of children with ASD and provide auditory-video feedback to calm down children with ASD. Schmidt et al. [17] designed a spherical video-based virtual reality (SVVR) mobile application to facilitate social skill training for individuals with ASD. Their study evaluated user experience, usability of application, and utilization feasibility, and found an overall positive user experience. On the other hand, robot-assisted therapies were commonly seen in emerging ASD studies. Robots were always built with various sensors, through which robots could get the necessary signals and interact as a human-like friend with ASD children [18]. For example, an NAO robot was a programmable robot equipped with cameras, microphones, and tactile sensors [19]. In a recent study conducted by Ali et al. [20], the NAO robot was programmed to provide visual, auditory, and tactile stimuli to engage children with ASD. Its camera monitored and recorded children’s joint attention. KASPAR is another child-sized robot equipped with tactile sensors. Costa et al. [21] used the KASPAR robot to help teach children with ASD appropriate tactile interactions. However, these robots did not have body-worn sensors, so children’s atypical sensory responses were manually analyzed based on observations in previous robotic studies.

It has been found that previous studies have used a range of methods, including a sensory profiling questionnaire, sensors, machine learning techniques, or IoT, to measure sensory responses and associated behavioral outputs, implying the feasibility of combining several off-the-shelf technologies to obtain comprehensive information for the development of a real-time monitoring system for children with ASD. Therefore, the goal of this study was to design a sensory management recommendation system (hereinafter referred as SMRS) specifically for children with ASD, using phone devices and sensors. The SMRS could understand a user’s sensory processing pattern, record and analyze sensory input, and act as a ‘specialist’ companion to recommend sensory management strategies in real time. The SMRS was implemented on a mobile phone because the growing usage of phones and sensors provided low-cost platforms for people to track information and access support [22]. Furthermore, it was suggested that the stigma associated with the disability could be reduced by using mobile phones, given that mobile phones could successfully address the challenges of public exposure [22,23]. The affordability and portability of the SMRS may make it easier for families of ASD children to access, in particular for those from relatively low-income countries. The SMRS in this study was evolved from the prototype which was tested successfully in researchers’ previous pilot study [4]. This study extended the previous paper by describing the full infrastructure of the SMRS and evaluating its effectiveness.

## 2. Materials and Methods

### 2.1. System Infrastructure

The proposed sensory management recommendation system (SMRS) was capable of collecting a user’s (child with ASD) sensory profile and allowing real-time data acquisition from the surroundings of the user through sensors. Firstly, the caregiver of the user should sign up for an account and complete a Sensory Profile of Children Three to Ten Years Caregiver Questionnaire [24], which is a standard sensory profiling tool that assesses the sensory processing pattern of a child. This questionnaire could elicit children’s sensory preferences or limitations, and classify their sensory pattern under four quadrants based on Dunn’s model of sensory processing [25]. Table 1 described the characteristics of individuals under the four quadrants. The measuring module of the SMRS collected physiological data such as skin conductivity, heart rate, and hand movements, and environmental data such as temperature, humidity, light exposure, and noise. The recording frequency was 1 hertz. Moreover, on the basis of acquired sensory information and machine learning models, the SMRS predicted the level of stress and attention of the child. The real-time data and predicted stress and attention level were further transmitted to the cloud server embedded with fuzzy logic controllers for sensory management strategy making. If the analyzed outcome was ‘High Risk’ (atypical sensory responses situation) and a sensory management recommendation was given, the alert could be initiated automatically by the SMRS. The message subscription allowed an alert to be sent to the caregivers or teachers via Short Message Service (SMS) regarding the occurrence of atypical situation (which was identified as ‘High Risk’ by fuzzy logic controllers) so that they could provide an intervention as suggested by the SMRS timely. The overall system infrastructure and working flow were presented in Figure 1.

### 2.2. Sensor Fusion and Data Management

To achieve proposed functions, the design of SMRS consisted of sensing devices, an iOS-based software application, and a cloud server for data management. iPhone [26], Apple Watch [27], and Arduino sensors [28] were the sensing devices used to record environmental and physiological data. The sensors and microcontrollers used in the SMRS are listed in Table 2, along with the purpose of each component. Figure 2a presents the circuit diagram of the Arduino board and Figure 2b is a prototype of the Arduino board. More technical details are presented as guidelines for sensor fusion and data management in the Appendix A.

An iOS-based application was created to connect the Arduino UNO and gain access to the built-in sensors on the iPhone and Apple Watch. The application allowed the caregivers to select a language they preferred (Figure 3a), to complete the sensory profile caregiver questionnaire (Figure 3b), and to view the environmental/physiological changes around the ASD user (Figure 3c) and sensory management strategies.

The cloud server for the SMRS was equipped with an Intel Xeon Processor, 4 GB RAM, and a Linux Ubuntu Operating System, which could handle multiple threads in parallel to receive data from different devices. The fuzzy logic controllers deployed on the cloud server could process the data received and return outputs to the SMRS. When the SMRS completed a recording session, sensor data were packed up with personal information (e.g., gender, date of birth, sensory profile) and sent to the cloud server so that researchers could access and manage the data accordingly.

### 2.3. Data Acquisition and Machine Learning Model Training

To acquire a training set of data with indicative labels for attention and stress detection, this study firstly used the SMRS prototype to record data in some children with ASD at a testing room of a rehabilitation center for several months.

Data acquisition was conducted with 35 children with ASD (mean age: 5.3; 29 boys, 6 girls, gender ratio: 4.83:1). Each child was told to complete 15 data collection sessions in different settings where environmental influences (i.e., temperature, noise, and brightness) were controlled during the session. In each session, three classical attention tasks (i.e., counting, picture matching, and drawing) for ASD children were given with audio instructions. Children’s performances on tasks were rated by an accuracy score from 0 (0% correct) to 1 (100% correct). Detailed descriptions of these procedures can be found in [4]. Finally, 521 valid datasets were obtained. Each dataset recorded sensor data during one session for approximately 15 min at a frequency of 1 hertz. Two healthcare professionals (qualified ASD specialists) were invited as assessors to classify children’s attention and stress levels at the end of tasks. They labelled the attention level of children as low or normal. Stress levels were labelled as low, moderate, and high. Since the data acquisition was long-term work, two healthcare professionals were not always available during that period. They successfully labelled 222 datasets. By combining task accuracy scores with classifications from professionals, researchers calculated the entropy and information gain [29] of the labelled sample to determine the best split points on attention and stress for all the 521 datasets. The calculation results showed that when employing the following classification, the data had the highest information gain values, indicating the best quality of the splits. A task accuracy score higher than 0.6 was considered to be within a normal level of attention, indicating that the child could generally pay attention to the task. A task accuracy score less than or equal to 0.6 was classified as the inverse of normal, indicating a low level of attention. Children relaxing in moderate environmental conditions were classified as having low stress, whereas performing tasks in moderate and extreme environmental conditions were classified as creating moderate stress and high stress, respectively. Features extracted as predictors for attention and stress detection were provided in Table 3.

Several machine learning models were implemented and compared for attention and stress detection. The investigated models included Logic Regression (LR), K Nearest Neighbor (KNN), Random Forest (RF), Artificial Neural Network (ANN), and Gradient Boosting Decision Tree (GBDT). The pre-processed data were split into the training and testing datasets by adopting 80:20 as the ratio of training:testing dataset. Five-fold cross-validation was performed with Grid Search on the training dataset to prevent overfitting and to obtain the best parameters for LR, KNN, RF, and GBDT. The best parameters evaluated by the mean cross-validation scores were used for the final model training. For ANN, loss functions with L2 regularization were used to optimize the model [30]. The performance of each model on the testing dataset was evaluated by accuracy and F1-score, which were computed by the following equations:(1)$$Accuracy=\frac{\mathrm{number}\;\mathrm{of}\;\mathrm{correct}\;\mathrm{predictions}}{\mathrm{total}\;\mathrm{number}\;\mathrm{of}\;\mathrm{predictions}\;\mathrm{made}},$$
(2)$$F1-score=2\times \frac{\mathrm{precision}\times \mathrm{recall}}{\mathrm{precision}+\mathrm{recall}},$$
(3)$$Precision=\frac{\mathrm{true}\;\mathrm{positive}}{\mathrm{true}\;\mathrm{positive}+\mathrm{false}\;\mathrm{positive}},$$
(4)$$Recall=\frac{\mathrm{true}\;\mathrm{positive}}{\mathrm{true}\;\mathrm{positive}+\mathrm{false}\;\mathrm{negative}}.$$

For attention detection, which was a binary classification, a weighted F1-score was calculated, while a macro F1-score was computed for stress detection, which was a multi-class problem [31]. Considering the machine learning model needed to be implemented on the mobile phone devices, the response time of a model was another factor that researchers investigated. All the models were processed on a laptop CPU and the inference time of each model was calculated and compared:(5)$$Inference\;time=\frac{\mathrm{total}\;\mathrm{time}\;\mathrm{taken}\;\mathrm{to}\;\mathrm{calculate}\;\mathrm{the}\;\mathrm{outputs}}{\mathrm{number}\;\mathrm{of}\;\mathrm{samples}}.$$

The accuracy, F1-score, and inference time on the testing data set of LR, KNN, RF, ANN, and GBDT models with optimal hyperparameters for stress and attention detection are presented in Table 4. It could be noticed that GBDT significantly outperformed the other models on attention detection with the highest accuracy (86.67%) and F1-score (0.8772). Machine learning models had overall better performance on stress detection than attention detection. The accuracies of RF, ANN, and GBDT models on stress detection were over 95%. RF had a higher accuracy (98.82%) and F1-score (0.9851) than ANN and GBDT. Most models could process an input within 0.1 millisecond (ms). Finally, two ensemble learning models, GBDT and RF, with the highest accuracy and generally short inference time, were chosen to be embedded into the SMRS for attention and stress detection, respectively [4].

### 2.4. Sensory Management Strategy Making

Real-time data collected by sensor fusion and predicted outcomes were further processed through the sensory management strategy making module where fuzzy logic controllers were implemented. The study employed fuzzy logic because it is a very classical and easily implemented method that imitates the human strategy-making mechanism [32]. It took the best decision for the given conditions based on some set of rules [33]. Before the fuzzy logic controllers could be used for the SMRS, all the inputs and fuzzy logic rules must be predefined. Therefore, researchers firstly gathered information about sensory management strategies through focus group consultations with 10 ASD specialists, combining with knowledge from ASD sensory toolkits [34,35]. Key environmental conditions that may trigger sensory management strategies included loud noises, bright or dark lights, and warm or cold room temperatures. As suggested by the focus group, the length of time that atypical sensory responses lasted was a factor that should be considered. The strategy for long-term and short-term atypical sensory responses could be different given that children with ASD have some degree of self-management ability. Secondly, in order to obtain more validated combinations of fuzzy logic rules and outcomes, researchers conducted a survey with 242 ASD specialists. Scenarios with risky conditions were shortlisted in the survey. Sensory management strategies obtained from focus group consultations and toolkits were listed as options. Survey results were finally interpreted into 63 fuzzy logic rules. Three independent fuzzy logic controllers were designed to process brightness, temperature, and noise stimuli in parallel. Each fuzzy logic controller contained 21 rules. A full list of rules is attached in Appendix B. Inputs to the fuzzy logic controllers included sensory stimuli (temperature/noise/brightness), duration of atypical sensory responses, and attention and stress levels. Trapezoidal and Gaussian membership functions were used to fuzzify the inputs. Figure 4 depicted the membership functions of the inputs and output.

The selection of the defuzzification method usually influenced overall performance of the fuzzy logic controllers [36]. This study opted for Largest of Maximum (LOM) method because it was more suitable for the general design of the fuzzy logic controllers. Researchers compared it with another most popular defuzzification method, which was Centroid method [37]. Two methods were tested on 21 different combinations of inputs. Centroid method only returned 79.4% of outcomes as expected in the testing because it usually led to a reasonable control action. To simplify, if there were two rules: ‘IF Temperature is High AND Duration is Short, THEN Outcome is Low Risk’, ‘IF Temperature is High AND Duration is Long, THEN Outcome is High Risk’, when the temperature was high, and the duration of atypical sensory responses was approaching long, the Centroid method averaged the two possible outputs to get the unwanted result ‘Medium Risk’. In this study, it was more important to detect ‘High Risk’ accurately. Therefore, the LOM method, which selected the largest output value whose membership value reached the maximum [38], was used in this study. Expectedly, the LOM method yielded superior results by returning all outcomes accurately in the testing. Table 5 provides some examples of the tested inputs, outcomes based on the LOM method, and strategies recommended by the SMRS.

Fuzzy logic controllers were coded by the Python language and deployed on the cloud server. The SMRS application on the phone was able to receive the outcomes of fuzzy logic controllers. When the outcome was ‘Low Risk’, the SMRS would show ‘No impact’ (Figure 5a). When the outcome approached ‘High Risk’, then the SMRS identified the triggering sensory input and recommended a proper strategy. The SMRS allowed an automatic message alert of a ‘High Risk’ situation. The recommended strategy could be sent to a corresponding caregiver or teacher via SMS message if they activate the automatic message alert by entering their phone number (Figure 5b).

### 2.5. Real-Life System Evaluation

The SMRS Beta App has been released on TestFlight for real-life system evaluation since March 2022. TestFlight is Apple’s beta testing service with which developers can invite testers simply by sharing a public link [39]. In this study, the evaluation of SMRS measured how accurately the SMRS identified the abnormal attention and stress levels of the children with ASD in real-life cases of different conditions of sound, light, and temperature. Separately, the study investigated the effectiveness of management strategies recommended by the SMRS on children’s performance improvement. Furthermore, the caregivers’ level of satisfaction in terms of system utilization, such as user interface, intention of long-term use, were assessed as well. ‘Real-life’ here referred to the regular environment in daily life in which there was no preliminary control on the generation of sound, visual, or tactile-related stimuli unless the SMRS recommended them to make an adjustment.

The evaluation was performed on 30 preschool children formally diagnosed as ASD. Another 30 gender- and age-matched TD children were involved as well for comparison. Table 6 highlights the demographic characteristics of participants. As shown in Table 6, the testing sites included several childcare centers in Zhejiang Province of China. The SMRS were tested across sites to demonstrate that the application and results of the SMRS intervention were not limited to a specific place.

The evaluation sessions were conducted in normal classrooms equipped with a desk, chairs, and necessary facilities (such as Figure 6a). The Arduino board was placed near the participant. An Apple Watch and GSR sensor were worn by the participant (Figure 6b). As shown in Figure 6c, in each testing room, there were one teacher and one participant at a time, with the caregiver using the SMRS and observing around the corner.

Before the formal evaluation sessions, caregivers were invited to install the SMRS Beta App and sign up an account for their children. If the caregivers did not own an iPhone or Apple Watch, test iOS devices that belonged to the research team were lent to them. All the participants and corresponding stakeholders (i.e., caregivers and teachers) were given coaching sessions about testing settings in advance. Each participant should undergo three sessions: no-SMRS session, SMRS session 1, and SMRS session 2 as described in Section 2.5.1, Section 2.5.2 and Section 2.5.3. Duration of each individual session was 30 min. Any two sessions for each participant should not be scheduled in one day to avoid a possible short-term effect such as fatigue and stress.

#### 2.5.1. No-SMRS Session

Prior to the first testing session with SMRS, the classroom teacher and caregiver provided a baseline rating regarding the child’s attention and stress in a no-SMRS condition. The child took a class as normal. After the session, the teacher and caregiver rated the child’s performance by using a report form adapted from the Caregiver-Teacher Report Form for Ages 11⁄2-5 (C-TRF) [40]. C-TRF is a well-validated instrument which evaluates behavior problems that occur in the classroom across multiple domains including anxiety, stress, attention, and social interaction [41]. Each item on the problem section of the C-TRF contains a statement about a child’s behavior. Response choices include: ‘Not True’ (scored as 0), ‘Somewhat or Sometimes True’ (scored as 1), and ‘Very True or Often True’ (scored as 2). The adapted C-TRF in this study includes items listed in ‘Anxious/Depressed’ and ‘Attention Problem’ subcategories. The adapted version of C-TRF was attached in the Appendix C.

#### 2.5.2. SMRS Session 1

In the first SMRS session, children and caregivers used the SMRS in the same class as the No-SMRS session. The researchers helped the child put the watch and GSR sensor on before the class started. The caregiver held the phone and used the SMRS on the phone in the classroom. If the detection results regarding attention or stress were wrong, the caregiver should make a real-time correction on the anomaly detection by clicking the ‘correct’ button and providing true labels on the phone (Figure 7). In this session, the caregiver or teacher would not follow immediate strategies recommended by the system for attracting the child’s attention or calming down the child. After the session, the caregiver and teacher completed the adapted C-TRF. Results from this session were used to evaluate how accurately the SMRS identifies the abnormal attention and stress levels of the children with ASD. By comparing the reported scores between the no-SMRS session and SMRS session 1, the researchers could discuss whether the implementation of wearable devices would influence children’s attention and stress in the classroom or not.

#### 2.5.3. SMRS Session 2

The procedures described for the testing preparation in the SMRS session 1 were identical for the SMRS session 2. However, during this session, when the SMRS identified an abnormal situation and the system generated a recommended strategy, the teacher in the classroom would simultaneously receive a text message of the recommended strategy. The teacher should take actions quickly as instructed by the strategy, such as using deep pressure (Figure 8a), fidget toys (Figure 8b), or playing a calming video (Figure 8c), to help the child pay attention or calm down during the class. Similarly, after the session, the caregiver and teacher completed the adapted C-TRF. By comparing the reported scores between SMRS session 1 and SMRS session 2, the researchers could investigate whether the management strategies recommended by the SMRS were helpful on children performance improvement. If no alerts happened when conducting sessions with a child from ASD group, the child would be asked to go through SMRS sessions again with the caregiver’s consent.

#### 2.5.4. Post-Session Evaluation

Following the completion of the testing, caregivers of the participants were invited to evaluate the overall functionality of SMRS by completing a System Usability Scale (SUS) questionnaire [42]. The SUS is a 10-item standardized questionnaire designed to measure users’ perceived usability and satisfaction of a system. As shown in the Appendix D, statements arranged as odd numbers are positively expressed and statements with even numbers are negatively expressed. Responses of each statement range from ‘Strongly Disagree’ to ‘Strongly Agree’ on a 5-point Likert scale.

### 2.6. Ethics Statement and Material Interpretation

The research protocol of this study was reviewed and approved by the Research Ethics Committee of the University of Nottingham Ningbo China. Informed consent was obtained from all participants involved in the study. For children, consent must have been given by their caregivers before participation. Caregivers had the right to withdraw their children from the study at any time.

The materials used in this study, including software application, questionnaires, research protocol, and participant information sheet and consent form, were prepared in English initially. However, participants in this study were living in China and using Chinese. The validated Chinese versions of sensory profile questionnaire, C-TRF and SUS questionnaire were used in this study. Other materials which did not have official Chinese versions were interpreted into Chinese by the first author who comes from China. The results and materials were presented in English in this study.

## 3. Results

This study utilized affordable sensors and off-the-shelf mobile devices to achieve proposed measurement, analysis, and strategy making. In all the testing cases, users’ sensory profile, physiological data, and information about user environment were successfully recorded by the SMRS. Machine learning models integrated in the SMRS could compute real-time user attention and stress using the collected sensory information. Fuzzy logic controllers that were deployed on the cloud server could generate fuzzy outcomes, which were responsible for activating the alert and making sensory management recommendations.

In the evaluation study, all the ASD and TD participants completed the required three test sessions. Each caregiver observed their child’s performance in each of the three sessions. The duration of every individual session was set to 30 min by researchers. During each session, a caregiver’s real-time reports on incorrect predictions through an interface in Figure 7 were interpreted into the number of wrong prediction cases. The wrong prediction cases were averaged for both groups of participants to give an indication of, overall, how many false predictions were made by machine learning algorithms in 30 min. Average wrong prediction cases, average ratings from C-TRF of ASD and TD groups, and standard deviations (SDs) are presented in Table 7.

### 3.1. Detection Accuracy

When examining wrong prediction cases, researchers identified that the ASD group reported more wrong prediction cases on attention than stress in the real-life situation. This meant that the accuracy of the attention model was not as satisfactory as the stress model in real-life practice, consistent with the results of model training in Section 2.3. The TD group obtained lower average ratings of C-TRF on both attention and stress domains, indicating that TD children might have a better ability of attention and stress management than the ASD group. TD children were more likely to make their attention or stress stable at a moderate level, with a higher tolerance to an unfriendly environment. The attention model could detect most of the state of TD children correctly as well. However, caregivers of TD children reported more wrong prediction cases of stress than those of ASD children. One reason for this could be that the data used for machine learning training were all from ASD children. Inputs corresponding to ‘uncomfortable level’ for an ASD child might be still within a TD child’s ‘comfort zone’, making the machine learning model generate wrong predictions for TD children.

By comparing the detection accuracy between SMRS session 1 and session 2, the number of wrong prediction cases dropped in session 2 where teachers implemented strategies to adjust the environment and help the children. It was suggested that machine learning models embedded in the SMRS were more usable for ASD children and had better performance in a comfortable environment. Admittedly, prediction accuracy of the attention model needs further improvement.

### 3.2. Effectiveness of the SMRS Intervention

To investigate the effectiveness of the SMRS intervention on children’s performance improvement, the *t*-test was employed to check if differences existed between the no-SMRS session and the SMRS session 2 for two groups. The magnitude of the differences between the no-SMRS session and SMRS session 2 was examined by calculating the effect size (d). The effect size was computed by dividing the mean change by standard deviation between two sessions. Cohen [43] labeled an effect size ‘small’ if d ≥ 0.2 and < 0.5, ‘moderate’ if d ≥ 0.5 and < 0.8, or ‘large’ if d ≥ 0.8.

A summary of the *t*-test and effect sizes are presented in Table 8. The analyses for each rating score given by caregivers and teachers on the C-TRF revealed significant performance differences in the ASD group between the no-SMRS session and the SMRS session 2 (*p* < 0.001). It indicated that the use of SMRS and the application of strategies recommended by the SMRS could help improve ASD children’s attention and reduce stress. Although differences in the TD group were not significant on the C-TRF rating of attention, caregivers also observed reduced stress in TD children (*p* < 0.05). Overall, *t*-test results suggested the positive impact of the SMRS intervention on sensory management in children with ASD. However, another index—effect size—was found only to be moderate for attention improvement and small for stress relief.

### 3.3. Level of Satisfaction in Terms of System Utilization

Caregivers’ perceived satisfaction in terms of system utilization was measured using the SUS. The average mean score for the 10 SUS items are presented in Table 9. Caregivers’ rating for the 6th statement was all scored below 2, suggesting that they generally disagreed that there was too much inconsistency in the system. They also did not perceive the system to be cumbersome to use or unnecessarily complex, given that the scores of the 2nd and 8th statement were low. Mean scores relating to the statement ‘I found the various functions in this system were well integrated’ were particularly high with similar standard deviations for both ASD and TD groups. However, it was noticeable that the statement ‘I think that I would need the support of a technical person to be able to use this system’ also obtained high scores, indicating that some instruction and assistance were required by the user group before they were able to use the system themselves.

The overall SUS scores in Table 9 were calculated with reference to the practical guidance developed by Sauro [44]. The scores for each statement were interpreted to a new number on a normalized scale of 0–4, summed, and then multiplied by 2.5. Generally, a SUS score above 68 was considered to have above-average usability. The mean SUS score of the ASD group and the TD group was 70.5 and 68.3, respectively, over the average SUS rating of 68. According to the practical guideline on the interpretation of a SUS score [44], a score above 70 suggested that the user-friendliness was good.

## 4. Discussion

The evaluation results suggested the benefit of the SMRS for preschool ASD children in real-life classroom settings. The SMRS could provide overall correct predictions on the attention and stress levels of children with ASD, identifying distractions and anxious situations. Statistical analysis revealed that the application of strategies recommended by the SMRS had a positive impact on ASD children’s sensory management in class, improving their attention level and reducing stress. The results of SUS survey suggested that caregivers of ASD children contended that the SMRS was user-friendly and various functions, such as real-time monitoring, detection, alerts, and strategy making, were well integrated.

The utilization of the SMRS in this study revealed that such a sensor and machine learning-based system could work as an efficient ‘specialist’ companion in real-life classroom settings for ASD children. The SMRS sensed ASD children’s environment, detected their attention and stress, and provided sensory management strategies to help mediate the negative affect of an unfriendly environment.

### 4.1. Comparisons with Existing Sensory-Based Technologies

The researchers compared the SMRS with some existing sensory-based technologies for ASD children. Related work referred to in the Introduction section targeting sensory issues of ASD individual were compared with the SMRS. As there were no previous studies that utilized the exact same materials and methods, comparisons mainly focused on the system features and methodological quality.

As shown in Table 10, it can be found that most technologies take advantage of sensors for monitoring. However, many of them fail to profile users’ sensory processing patterns to make recommendations on management strategies to help users. There is evidence to suggest that sensory processing patterns are idiosyncratic in individuals with ASD [25]. Researchers’ previous work compared prediction accuracy of machine learning models on attention detection before and after removing sensory profile features [4]. The results showed that the prediction accuracy of all models dropped dramatically after the sensory profile features were removed, which indicated that the inclusion of a sensory profile could help ensure the correctness of detection. Although many sensory profiling tools have been widely used in healthcare services, very few technologies have involved these tools in their design.

Regarding data analysis, references [16,20,21] still depended on an ASD expert’s manual analysis, which would require the continuous involvement of expert assistance. It complicated the use of technologies in daily life and increased the cost for ASD families. Five out of eight studies in Table 10 have used machine learning or cloud computing to enable on-device data analysis. Most of them were published in or after 2019, suggesting an emerging trend of studies executing computation directly in the system.

To date, to the researchers’ best knowledge, this is the first work that has combined a standardized sensory profiling tool, sensor monitoring, data analysis, and sensory management strategies in one low-cost system for supporting ASD families to deal with sensory issues. These features highlighted the novelty of the SMRS. Moreover, one challenge for many previous studies was to evaluate the intervention with a large sample of ASD individuals. Some studies only involved a few ASD participants in the evaluation or did not report evaluation. This study conducted an evaluation with a larger sample than previous studies, following a well-defined protocol aiming to make the results more generalizable.

### 4.2. Limitations and Recommendations for Further Study

The development of the SMRS complemented other existing research by providing a comprehensive sensor and machine learning-based monitoring system. The SMRS was available on TestFlight which could be installed after being invited. This allowed continuous test of the SMRS and data collection in children with ASD. With increased data size, machine learning models will be further trained to improve prediction accuracy. Researchers in this study further used data from 30 ASD participants in the evaluation study to train the GBDT model for attention detection. Experiments using an increased training sample witnessed an improvement of the performance of GBDT model with the detection accuracy increased from 86.67% to 92.61%. This implies that after a period of beta test involving more ASD users, the SMRS can have better performance in identifying the risk situations for children with ASD.

Considering the moderate effect size of the SMRS intervention in an ASD group whose sample size was still limited, further efforts could be made to expand the strategy knowledge base by adding more effective intervention strategies, and to involve more ASD participants in the evaluation.

Another limitation of this study related to the restricted age range and condition (ASD) of the users. The sensory profile questionnaire in the SMRS was for children aged between three and ten only. The age of children involved in this study ranged from three to five. However, sensory issues in ASD may last throughout an individual’s life. Using an adolescent/adult sensory profile and the replication of the study with other age groups would increase the utility of the SMRS. Sensory issues were not only observed in ASD but also in disorders such as Attention Deficit Hyperactive Disorder or Sensory Processing Disorder. The SMRS could also be tested for other sensory-related disorders in the future.

Moreover, researchers noticed a slight increase of C-TRF scores in SMRS session 1 compared to the no-SMRS session, suggesting that sensors that touched the body of the child might make the child stressed or more easily distracted. A further system update may try to minimize the influence of tactile defensiveness due to the touch of sensor. For example, non-invasive sensors could be used, such as Kinect or apparel-based sensors. The IoT nowadays creates opportunities for human-machine or machine-to-machine communications, in which way the ‘things’ can not only identify sensory events in their surroundings but also react autonomously without human intervention [16]. Future work could make use of IoT to create a smart environment to implement the environmental control and calming strategy automatically so that the SMRS will not require the continuous involvement of a human assistant.

## Figures and Tables

**Figure 1 sensors-22-05803-f001:**
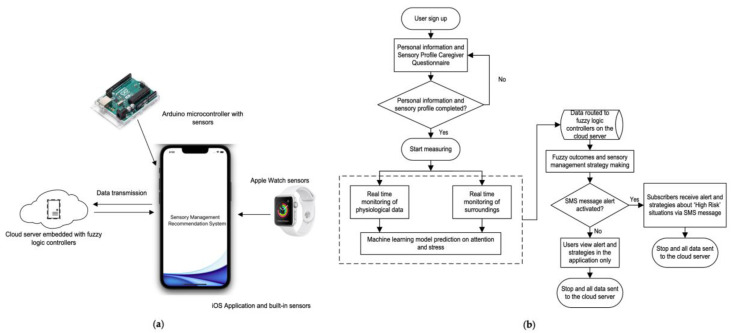
System infrastructure. (**a**) Data transmission network of the SMRS; (**b**) working flow of the SMRS.

**Figure 2 sensors-22-05803-f002:**
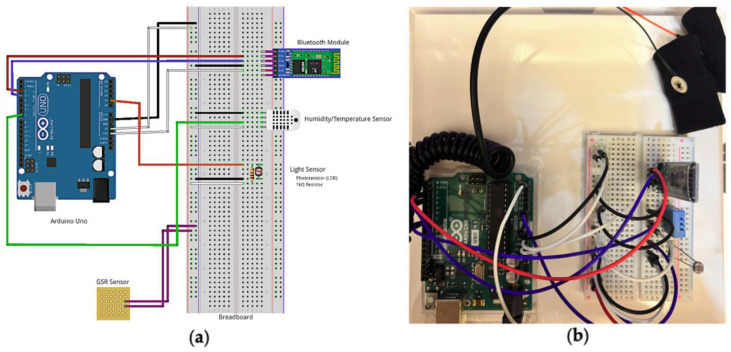
Arduino UNO circuit connection. (**a**) Circuit diagram; (**b**) prototype of the Arduino board.

**Figure 3 sensors-22-05803-f003:**
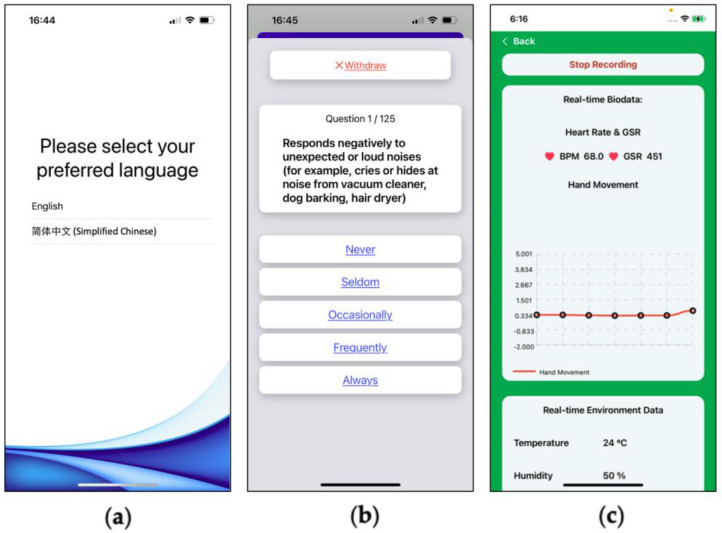
SMRS application interface. (**a**) Language selection; (**b**) sensory profile questionnaire; (**c**) sensor data visualization.

**Figure 4 sensors-22-05803-f004:**
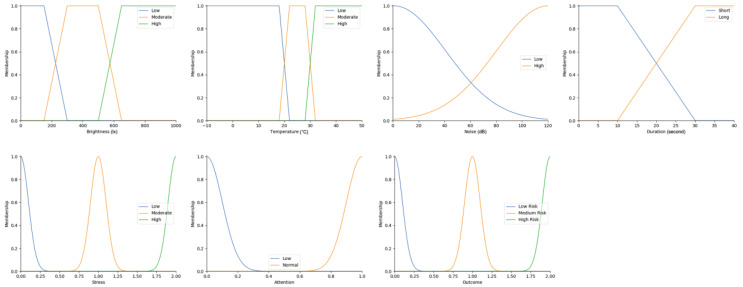
The graphs of the membership functions for the inputs and outcome.

**Figure 5 sensors-22-05803-f005:**
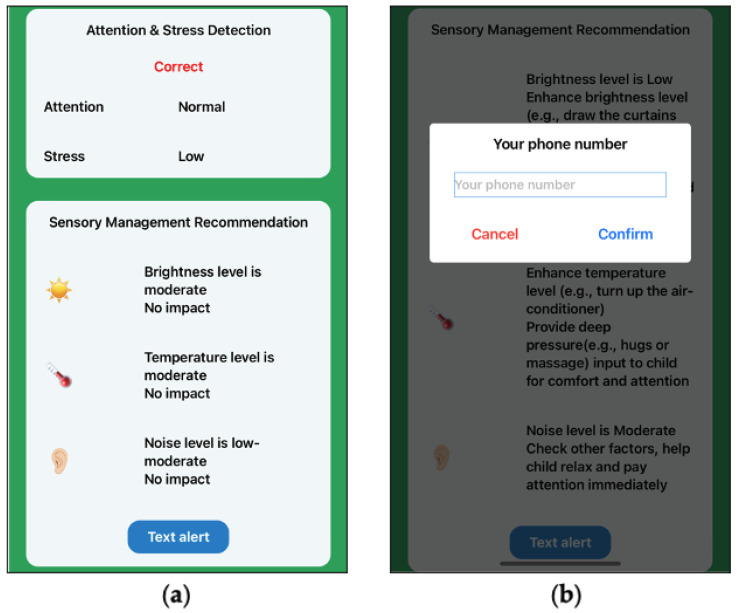
Interface of sensory management strategy recommendation. (**a**) Feedback interface; (**b**) message alert subscription.

**Figure 6 sensors-22-05803-f006:**
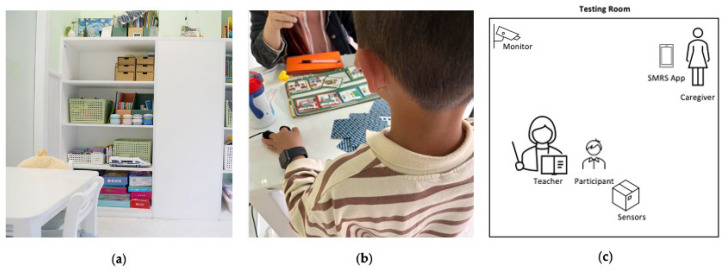
The classroom setup in the evaluation sessions. (**a**) A real-life classroom setting; (**b**) child worn sensors in the evaluation; (**c**) the setup for a SMRS session.

**Figure 7 sensors-22-05803-f007:**
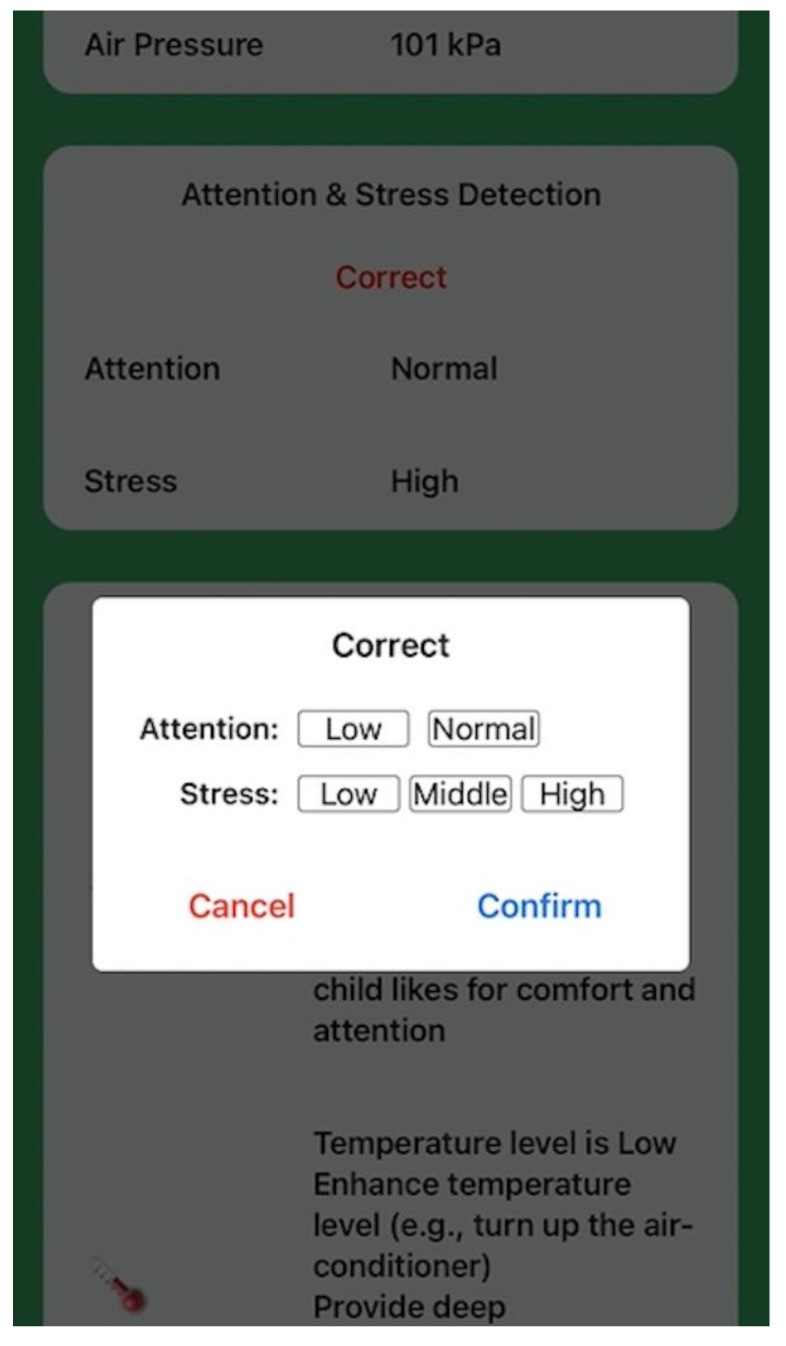
Caregiver feedback interface.

**Figure 8 sensors-22-05803-f008:**
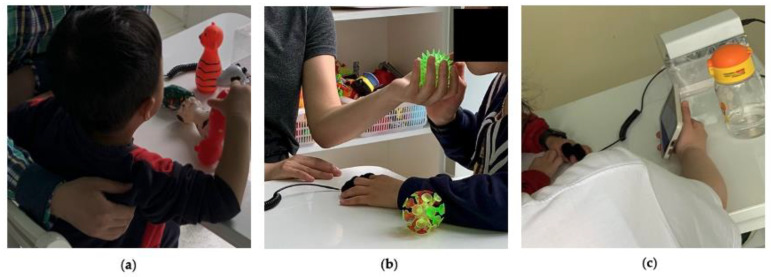
Teachers applied strategies recommended by the SMRS. (**a**) Deep pressure strategy; (**b**) fidget toy strategy; (**c**) calming video strategy.

**Table 1 sensors-22-05803-t001:** Characteristics of four sensory patterns [4].

Sensory Pattern	Characteristics
Low registration	Less likely to notice sensory input, may behave as passive or easy going.
Sensory seeking	Prone to add sensory events to daily life, may be very active or keep busy.
Sensory sensitivity	More likely to get distracted by sensory inputs, often show discomfort and sensitivity towards daily events.
Sensory avoiding	Prone to withdraw from overwhelming sensory stimulation, may be very ritualistic and rule-bound.

**Table 2 sensors-22-05803-t002:** Sensors and microcontrollers.

Sensor/Microcontroller	Unit	Purpose
Arduino UNO Rev3	N/A *	To fetch and transmit signal from sensors.
Apple Watch three-axis accelerometer	Sensor value	To identify the hand movements.
Apple Watch heart rate sensor	Beats per minute (bpm)	To measure heart rate.
DHT11 temperature and humidity sensor	Celsius (°C) for temperature, percentage (%) for humidity	To measure temperature and humidity level.
iPhone microphone	Decibel (dB)	To measure noise level.
iPhone barometer	Kilopascal (kPa)	To measure air pressure.
Light sensor (photoresistor)	Lux (lx)	To measure brightness level.
SEEED Grove Galvanic Skin Response (GSR) sensor	Sensor value	To detect skin conductivity.

* N/A: Not applicable.

**Table 3 sensors-22-05803-t003:** Extracted data features.

Category	Included Features
Environmental features	Temperature, noise, humidity, brightness, air pressure
Sensory profile features	Low registration, sensory seeking, sensory sensitivity, sensory avoiding
Physiological features	GSR, heart rate, watch accelerometer (mean absolute value of three axis)
Personal characteristics	Gender, age

**Table 4 sensors-22-05803-t004:** Model performance on attention and stress detection.

	Attention Detection			Stress Detection		
Model	Accuracy (%)	Weighted F1	Inference Time (ms)	Accuracy (%)	Macro F1	Inference Time (ms)
LR	65.71	0.6949	0.0052	65.30	0.5712	0.0013
KNN	81.90	0.8319	0.0291	93.92	0.9251	0.1041
RF	79.05	0.8000	0.0958	98.82	0.9851	0.0182
ANN	80.95	0.8246	0.0040	96.89	0.9592	0.0021
GBDT	86.67	0.8772	0.0046	98.50	0.9812	0.0366

**Table 5 sensors-22-05803-t005:** Outcome responses of fuzzy logic controllers and strategy making module on different combinations of inputs.

Inputs				Outcome Responses	
Sensory Stimuli	Attention	Stress	Duration (s)	Fuzzy Outcome	Recommended Strategy
Brightness = 100 lx	Low	High	25	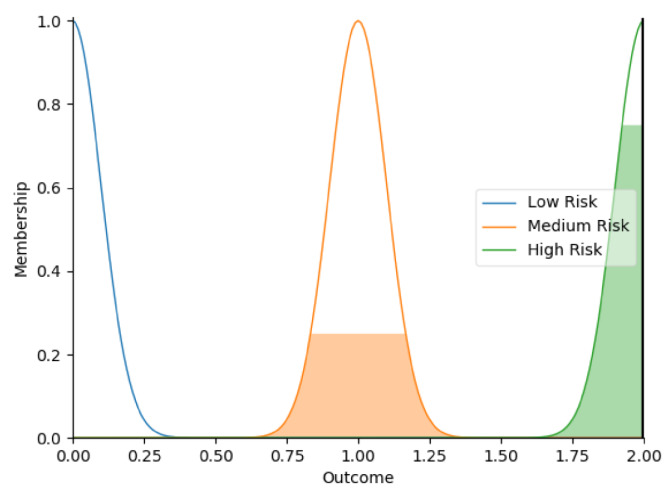	Brightness level is low. Enhance indoor brightness (e.g., draw the curtains open), use a phone to show pictures or videos that the child likes for comfort and attention.
Brightness = 400 lx	Normal	Low	40	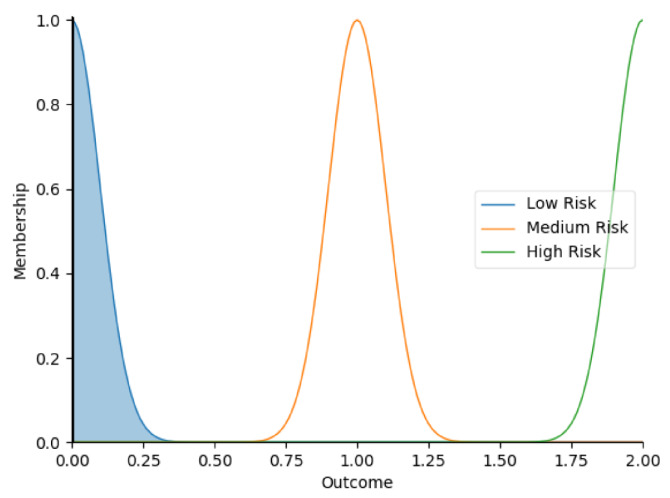	Brightness level is moderate. No impact.
Brightness = 750 lx	Normal	High	10	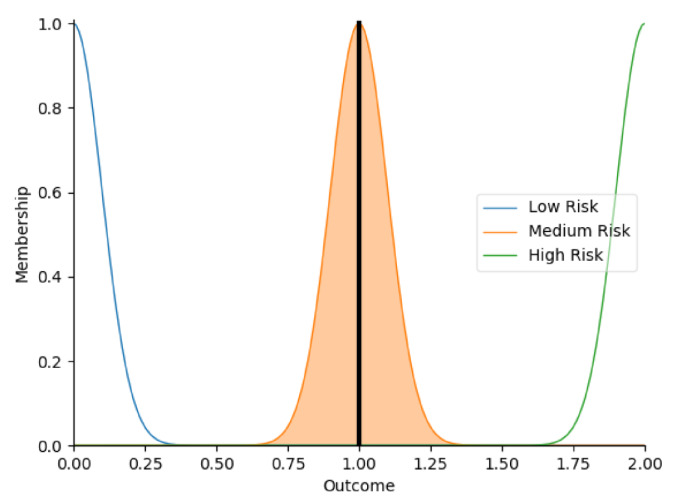	Brightness level is high. Reduce indoor brightness (e.g., draw the curtains). Keep observing.
Temperature = 15 °C	Normal	High	10	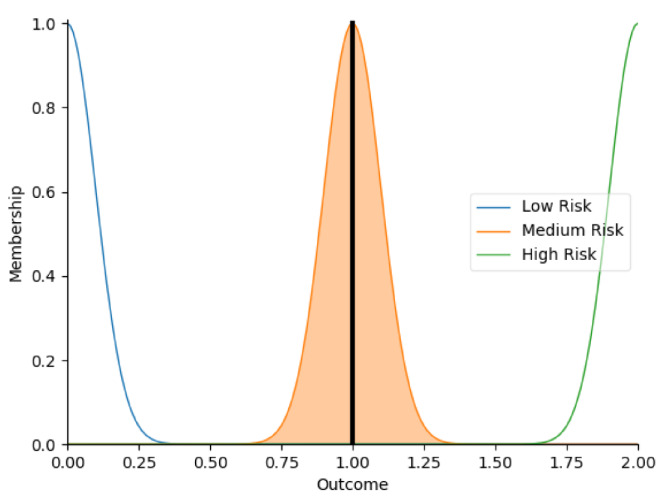	Temperature level is low. Enhance temperature level (e.g., turn up the air-conditioner). Keep observing.
Temperature = 26 °C	Normal	Moderate	25	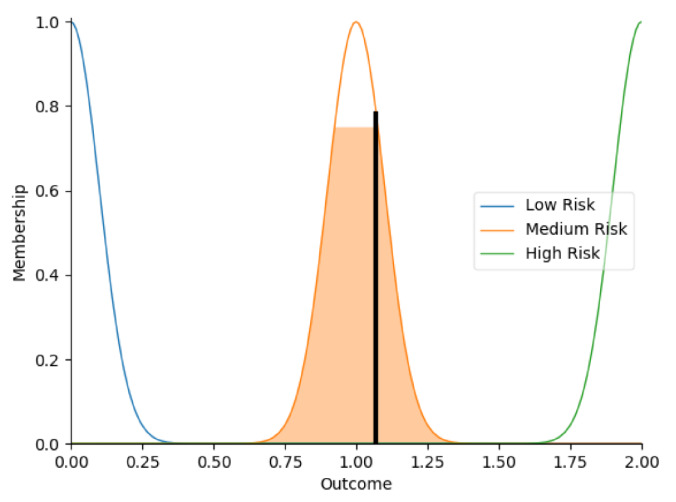	Temperature level is moderate. Keep observing.
Temperature = 32 °C	Low	High	40	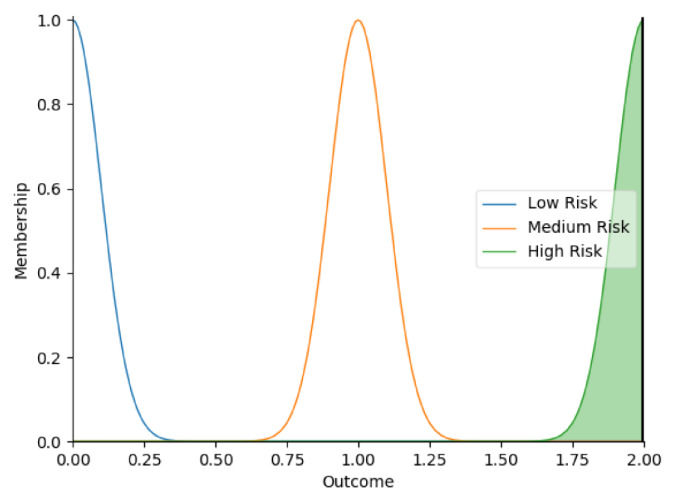	Temperature level is high. Reduce temperature level (e.g., turn on the fan). Provide some deep pressure (e.g., hugs or massage) input to child for comfort and attention.
Noise = 60 dB	Low	Low	25	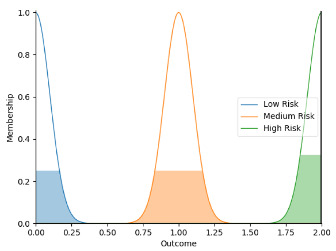	Noise level is moderate. Check other factors that may distract your child.
Noise = 70 dB	Normal	High	10	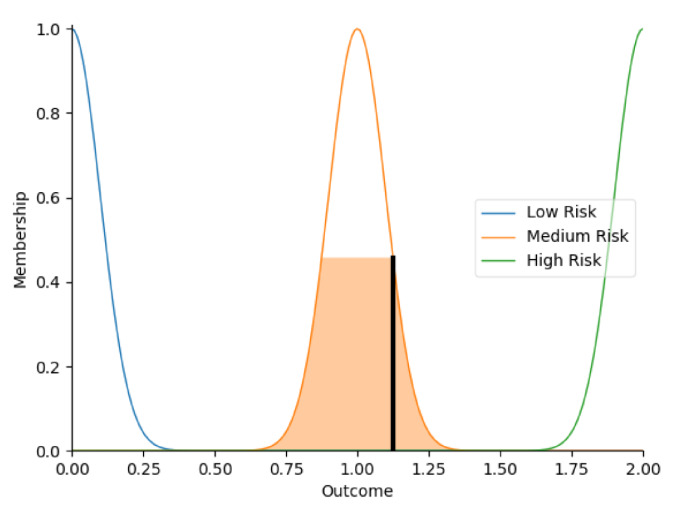	Noise level is moderate-high. Keep observing.
Noise = 80 dB	Low	Moderate	40	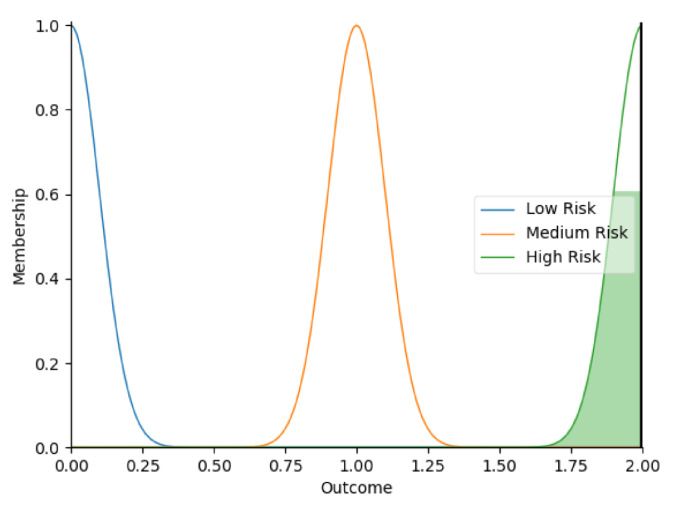	Noise level is high. Try to reduce loud (e.g., use noise-cancelling headphones or play calming music). Provide a fidget toy with texture that child likes for comfort and attention.

**Table 6 sensors-22-05803-t006:** Demographic information of participants of evaluation study.

Condition	Testing Site	Number of Participants	Average Age	Gender Ratio (Male:Female)
ASD	An ASD Rehabilitation Center in Wenzhou	15	4.3	12:3
	An ASD Rehabilitation Center in Ningbo	15	4.0	12:3
TD	A Public Kindergarten in Wenzhou	15	4.4	12:3
	A Private Childcare Center in Ningbo	15	4.3	12:3

**Table 7 sensors-22-05803-t007:** ASD and TD group data of different measures.

Group	Sample	Session	Rating Parameters: Mean (SD)					
			Wrong Prediction Cases— Attention	Wrong Prediction Cases—Stress	C-TRF Attention Score—Caregiver	C-TRF Attention Score—Teacher	C-TRF Stress Score—Caregiver	C-TRF Stress Score—Teacher
ASD	30	No-SMRS	/	/	8.1 (3.5)	8.6 (2.8)	4.3 (3.5)	4.9 (4.0)
		SMRS #1	21.9 (15.7)	6.7 (5.0)	8.3 (3.2)	8.7 (3.3)	4.4 (3.6)	4.9 (3.9)
		SMRS #2	11.0 (7.7)	4.2 (2.5)	6.5 (2.8)	7.0 (2.9)	3.4 (3.3)	3.7 (3.4)
TD	30	No-SMRS	/	/	1.6 (1.6)	2.0 (2.0)	1.6 (1.9)	1.5 (1.9)
		SMRS #1	5.2 (4.2)	18.6 (8.4)	1.8 (1.8)	2.2 (2.2)	1.7 (2.1)	1.7 (1.9)
		SMRS #2	4.8 (3.4)	13.3 (7.2)	1.5 (1.5)	1.8 (2.0)	1.2 (1.8)	1.3 (1.6)

**Table 8 sensors-22-05803-t008:** Summary of the *t*-test and effect sizes for both groups.

Measures	No-SMRS—SMRS Session 2					
	ASD			TD		
	t	Sig.* (2-Tailed)	d	t	Sig. (2-Tailed)	d
C-TRF Attention Score—Caregivers	4.732	<0.001	0.505	0.769	0.448	0.065
C-TRF Attention Score—Teachers	4.533	<0.001	0.561	1.229	0.229	0.100
C-TRF Stress Score—Caregivers	4.160	<0.001	0.265	3.340	0.002	0.216
C-TRF Stress Score—Teachers	5.288	<0.001	0.323	1.649	0.110	0.114

* Sig.: Significance.

**Table 9 sensors-22-05803-t009:** Caregivers’ SUS rating of the SMRS.

Statement	Mean Score (SD) and Range	
	ASD Group	TD Group
1. I think that I would like to use this system frequently.	3.87 (0.64); range: 3–5	3.67 (0.72); range: 3–5
2. I found the system unnecessarily complex.	1.67 (0.72); range: 1–3	2.07 (0.80); range: 1–3
3. I thought the system was easy to use.	3.67 (0.72); range: 3–5	3.73 (0.80); range: 3–5
4. I think that I would need the support of a technical person to be able to use this system.	3.73 (0.80); range: 3–5	3.93 (0.70); range: 3–5
5. I found the various functions in this system were well integrated.	4.20 (0.68); range: 3–5	4.33 (0.62); range: 3–5
6. I thought there was too much inconsistency in this system.	1.07 (0.26); range: 1–2	1.07 (0.26); range: 1–2
7. I would imagine that most people would learn to use this system very quickly.	3.53 (0.83); range: 2–5	3.40 (0.51); range: 3–4
8. I found the system very cumbersome to use.	1.13 (0.35); range: 1–2	1.20 (0.56); range: 1–3
9. I felt very confident using the system.	3.33 (0.90); range: 2–5	3.40 (0.74); range: 2–5
10. I needed to learn a lot of things before I could get going with this system.	2.80 (0.68); range: 2–4	2.93 (0.46); range: 2–4
Overall SUS Score (calculated as per [44])	70.5 (3.92); range: 62.5–80	68.3 (3.62); range: 62.5–77.5

**Table 10 sensors-22-05803-t010:** Comparison with other sensory-based technologies for ASD individuals.

Reference	Technology Features					Methodology Quality	
	Sensory Profiling	Physiological Monitoring	Environmental Monitoring	Data Analysis	Strategy Making	Evaluation	ASD Sample in the Evaluation
This study	Yes	Yes	Yes	Yes	Yes	Yes	30
[13]	Yes	No	Yes	Yes	Yes	Yes	20
[14]	No	Yes	No	Yes	No	Yes	20
[15]	No	Yes	No	Yes	No	Not reported	Not reported
[16]	No	Yes	Yes	No	No	Yes	1
[9]	No	No	Yes	Yes	Yes	Yes	10
[20]	No	Yes	No	No	No	Yes	12
[21]	No	Yes	No	No	No	Yes	8

## Data Availability

The data presented in this study are available on request from the corresponding author. The data are not publicly available due to ethical restrictions.

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
