# Peer review of "A Sensor and Machine Learning-Based Sensory Management Recommendation System for Children with Autism Spectrum Disordersâ€"

_sensors, 2022, doi:10.3390/s22155803_

Round 1
Reviewer 1 Report
The paper is well prepared and rather interesting. Some remarks about the deficiencies. Too many technical details, especially in chapter 2. If authors threaten them as necessary many details could be provided as an annex or supplementary material. These technical details aren't directly related to the scientific study provided. At the same time experiment conditions and methods needs to be described in more detail. E. g. big differences in the accuracy achieved when applying different machine learning methods to detecting attention and stress may be the result of overfitting (2.3) because a relatively not big data set was used in experiments. A more detailed presentation of the experiment will help to understand this (was cross-validation applied or not remained unclear from the presentation). When presenting the results of the caregivers survey not only mean but also standard deviation could be given: in this case, the standard deviation will give a better understanding of the spread of the values than the range. Comparison with the existing sensory-based technologies could be presented earlier since raises the question of why the authors first did the research and only later started to compare the proposed solution with other technologies. In principle, the presented study is bold from the scientific point of view.
Reviewer 2 Report
The manuscript describes a strategy to help children with ASD to manage stressful situations in a classroom environment. The strategy integrates, sensors with a questionnaire for a fuzzy logic based analysis for making recommendations to teachers and caregivers. The device and analysis side has not much novelty, however the real-time test results on 60 children that are presented could be valuable. Further discussion of the high "wrong prediction" numbers and potential improvement strategies is needed before publication. In its current form, the manuscript says, further improvement is needed but does not provide any further details. Also, it is not clear what is the meaning of wrong prediction cases? Is it per child? How many cases are there? Further explanation would be helpful.
